# Who does tracing work for? Characteristics of clients successfully re-engaged in ART care in sub-Saharan Africa after a tracing intervention: A systematic review

Anushka Reddy Marri[1], Allison Morgan[1], Mariet Benade[1,2,3], David B. Flynn[4], Mhairi Maskew[2], Nyasha Mutanda[2], Sydney Rosen[1,2]*

1 Department of Global Health, Boston University School of Public Health, Boston, Massachusetts, United States of America, 2 Health Economics and Epidemiology Research Office, University of the Witwatersrand Faculty of Health Sciences, Johannesburg, Gauteng, South Africa, 3 Department of Global Health, Amsterdam Institute for Global Health and Development, Amsterdam UMC, University of Amsterdam, Amsterdam, The Netherlands, 4 Department of Medical Sciences & Education, Boston University Chobanian & Avedisian School of Medicine, Boston, Massachusetts, United States of America

☺ These authors contributed equally to this work.
* sbrosen@bu.edu

## Abstract

Tracing HIV treatment clients who have interrupted or disengaged from care is a common, guideline-recommended practice globally. Most guidelines prioritize tracing based on clinical condition or HIV transmission risk, not likelihood of client traits that may affect return to care after tracing. Targeting tracing to those most likely to return could increase efficiency substantially. We conducted a systematic review to identify characteristics of clients most likely to return after tracing. We searched PubMed, EMBASE, and Web of Science for studies published between 1/2004 and 7/2025 that reported outcomes of tracing interventions in sub-Saharan Africa. Eligible studies reported characteristics of clients who interrupted care, were eligible for a tracing intervention with the intent to return them to care (i.e., not solely research to determine client outcomes after interruption) and were subsequently traced or had tracing attempted. Our primary outcome was client characteristics associated with return to care after tracing, compared to those who did not return after tracing or attempted tracing. We identified 13,208 articles; 9 met the inclusion criteria. Older age and female sex were the most consistent predictors of return after tracing. Earlier tracing (relative to last missed visit) was associated with return in 3 studies; 1 found the opposite. Frequent contact attempts, rural location, and psychosocial factors (stigma, disclosure) were also associated with return. Clinical characteristics (CD4 counts and WHO stage) showed mixed or null associations with tracing effectiveness. Characteristics of clients who return to care after tracing, compared to those who are traced or for whom tracing is attempted and do not return, are rarely reported, making it difficult to evaluate this intervention. Using a "high-benefit" approach to targeting

**Data availability statement:** All data reported are publicly available and we have indicated the fields that we have extracted in the points above. This paper is a systematic review and all input data are drawn from previously published sources. We have no additional data to share related to our paper.

**Funding:** Funding for the study was provided by the Gates Foundation through INV-031690 to Boston University. The funders had no role in study design, data collection and analysis, decision to publish, or preparation of the manuscript.

**Competing interests:** The authors have declared that no competing interests exist.

tracing—i.e., prioritizing based on likely benefit generated by a successful response, rather than clinical need—may potentially improve the efficiency of HIV programming.

## Introduction

In many sub-Saharan African countries, the successful scaleup of national HIV treatment programs has led to the achievement or near-achievement of global targets for the HIV response, known as "95-95-95" (95% of HIV-positive people know their status, 95% of those who know their status initiate treatment, and 95% of those on treatment are virally suppressed) [1]. Retaining ART clients in lifelong care, while maintaining high medication adherence to achieve viral suppression, is now one of the main challenges to HIV control [2,3]. Due to very high rates of interruption and disengagement from care--recently estimated at 23% and 47% by one year after initiation in South Africa [4] and Zambia [5], respectively--interventions that effectively return clients to care after interruption are a high priority.

One such strategy recommended by the World Health Organization (WHO) is client tracing [6]. Tracing (also referred to as tracking) utilizes health care workers, lay personnel, and mHealth technologies to contact and re-engage clients who have missed a visit or interrupted care. Tracing is generally conducted between one day and several months after an ART client has missed a scheduled clinic visit or medication refill. Typically the clinic at which the client is enrolled will i) observe a missed visit or medication refill; ii) place the individual in question on a tracing list or register; iii) have a nurse or a lay healthcare worker, such as a community health worker, counselor, or clerk, attempt to contact the individual by phone or text message if consent for contact has been provided; and iv) as resources allow, visit the individual's home if he or she cannot be reached by phone [6,7]. For many facilities, tracing has been the responsibility of an externally funded nongovernmental partner or community group or is only conducted when the facility has sufficient resources (cell phone airtime, fuel for vehicles, staff time, etc.) or when reporting to a government health agency or funder is required.

Evaluations of the effectiveness of tracing interventions in persuading disengaged clients to return to care have demonstrated mixed results. Success rates for tracing interventions vary between countries, strategies, and fidelity to tracing guidelines and implementation, with estimates of return rates (proportion of traced clients who return to care after tracing contact) ranging from as low as 31% to as high as 96% in the published literature [7–10]. Tracing interventions face many challenges, with efforts hindered by insufficient contact information, frequent relocation or migration among transient populations, and misclassification of clients' true engagement status [10,11]. A recent study in Tanzania, for example, reported that among 2,626 ART clients who were more than 90 days late for a visit, 31% were traced (of whom about three-fourths returned to care), while the rest had relocated (19%), transferred to a different clinic (11%), died (7%), or could not be contacted (32%) [12]. The process of tracing clients is also resource-intensive, costing as much as a year of treatment per client returned to care in the same Tanzanian study [12].

Due to high rates of interruption and disengagement and the scarcity of resources for tracing, both the WHO and many national guidelines have recommended client subgroups to prioritize for tracing. Prioritization is generally based on the client's current clinical condition and future risk of HIV disease progression or transmission, as shown in Panel 1.

**Panel 1. Examples of global and national guidelines for tracing procedures.**

| Country | Priorities for tracing | Recommended timing of tracing after missed interaction | Guideline recommended procedures |
|---|---|---|---|
| World Health Organization [6] | Initiated ART in the past six months with advanced HIV disease; abnormal lab results; deferred or declined treatment initiation; overdue for laboratory testing and consultation | >7 days | Rapidly trace clients who are 7 or more days late for a scheduled appointment using remote (phone, text), or in-person visits. Pair with peer outreach or economic support where feasible. Obtain consent at ART initiation and counselling, keep contact information updated, and train tracing staff in confidential, non-judgmental re-engagement. |
| Ghana [13] | Advanced HIV disease (AHD) | Not specified | Home visits and rapid tracing of AHD clients who missed appointments. |
| Kenya [14] | Client type (pregnant and breast-feeding women, child, comorbidity); proximity to clinic | 24 hours | Phone clients/caregivers within 24 hours of a missed appointment; if unsuccessful, call the treatment buddy. If still unresponsive after seven days and not confirmed as dead, transferred, or declining contact, initiate a home visit and refer untraced/non-returning cases to a multidisciplinary team for a further follow-up plan. |
| Malawi [15] | None indicated | 2 weeks | Attempt to contact the client or named guardian through phone calls or home visits (for consented clients) starting two weeks after a missed appointment until status is confirmed. An overdue client, not confirmed as stopped, deceased, or transferred, and out of ARVs for two or more months is classified as lost to follow-up/defaulted. |
| Nigeria [16] | AHD; clients missing TPT appointments | Not specified | People missing appointments should be actively tracked by phone or through home visits. |
| South Africa [17,18] | Early treatment period; AHD; abnormal test results; diagnosed but not started ART; overdue for assessment and/or investigation | 7 days | Trace clients who miss their appointment by more than five days through their preferred mode of contact (phone calls, SMS, or home visits) starting 7 calendar days after the missed appointment until status is confirmed, ensuring reintegration into care with additional psychosocial support when needed. |
| Uganda [19] | Initiating treatment with AHD in the past six months, with abnormal results, not initiating treatment, or overdue for clinical consults/laboratory tests | 7 days | Initiate remote outreach (phone calls, text messages, mail, email) and/or in-person tracing to locate disengaged clients for reengagement in care through supportive interventions such as linking to community-based follow-up services. Tracing should ideally begin for all clients who are seven or more calendar days late for a scheduled appointment. Clients with treatment interruption of 28 days or more should be classified as lost to follow-up and actively traced. |
| Zambia [20] | None indicated | 24 hours to 28 days | Conduct active client tracking through text messaging, phone calls, home visits, contact with community health worker, or contact with a treatment buddy/emergency contact starting within 24 hours of a missed pharmacy refill appointment and conducted at least six times using different modalities for up to 28 days, at which time the client should be designated as interrupted in treatment. |

AHD, advanced HIV disease; TPT, tuberculosis preventative therapy.

Tracing all clients lost to follow up or prioritizing clients based on worst clinical condition and highest risk of transmission ("high-risk approach") may appear efficient, as these clients would seem to suffer the most harm or cause the most harm to others from defaulting treatment [21]. However, studies in other disciplines are beginning to question this approach by instead asking which clients would generate the most benefit from an intervention ("high-benefit approach") [22]. While prioritizing clients for tracing based on clinical condition is intuitive from a risk minimization perspective, it does not consider the relative effectiveness of tracing among different subgroups of clients. There is no reason to believe that

individuals at greatest clinical risk are more likely to respond positively to a tracing intervention, for example, rather than those with some demographic, socioeconomic, geographic, and/or behavioral characteristic in common. As might be expected, individuals who are both at high clinical risk **and** have characteristics that make them prone to respond positively to tracing would probably be the most valuable subpopulation to target, if they could be identified.

To our knowledge, tracing has never been prioritized on the basis of its likely success with different subgroups (a high-benefit approach), rather than by individual clinical condition and individual risk of disease progression or transmission (a high-risk approach). If tracing is more likely to succeed with some subgroups than with others, the efficiency of tracing interventions could be improved substantially. As a starting point for reconsidering tracing prioritization, we conducted a systematic review of tracing studies with the goal of describing and comparing characteristics of clients who return to care as a result of tracing interventions to those who do not return to care and exploring factors associated with tracing success and failure.

## Methods

We conducted a systematic review of the peer-reviewed literature and published conference abstracts that reported on outcomes of tracing interventions in sub-Saharan Africa. The review was registered on the International Prospective Register of Systematic Reviews (PROSPERO) (CRD42024534323) (S1 Text). We report our findings in accordance with the Preferred Reporting Items for Systematic Reviews and Meta-Analyses (PRISMA) 2020 guidelines [23]. We note that although "disengaged" is becoming the preferred current term for describing ART clients who have discontinued care, most of the literature during our study period referred to such clients as "lost to follow up" (LTFU). In this review, we use the terms from the original publications when commenting on specific papers and "disengaged" for aggregate results [24].

### Proposed reporting framework

To understand characteristics of clients who are responsive to tracing efforts and ultimately re-engage in care, we devised a standard reporting framework (Fig 1). We propose that to better understand tracing successes, demographic and clinical characteristics of clients should be reported at each level of this cascade whenever possible, on the premise that systematic reporting would standardize outcome documentation across diverse settings, improve the quality and consistency of client data repositories, support evidence-based planning, and inform the development of more universal program strategies. This framework allowed us to search for and analyze sources which report differences between the last two boxes [4a and 4b], "Reached and returned to care" and "Reached and not returned to care," which are the groups of interest for this systematic review, and to look for differences between clients in boxes 3a and 3b, which indicate whether or not a client eligible for tracing could be contacted. We note that additional information should be added to this framework when it is used for reporting results, such as the number of times providers attempted to contact the clients in row 3 or the number of days until return in row 4. To illustrate how this framework can be applied in practice, we reference Fig 1 box codes (e.g., "Reached and returned to care" [Fig 1, box 4a]) in the manuscript where relevant.

### Search strategy, eligibility criteria, screening, and data extraction

Searches were conducted using PubMed, EMBASE, and Web of Science. The first search was conducted in April 2024, to identify studies published after January 1, 2004. The search was updated in July 2025 to identify published work from the previous year. Search terms developed by AJM, SR, and DBF combined Medical Subject Headings (MeSH) and free-text terms using Boolean operators (AND, OR) to refine results [25]. While the keywords remained consistent, the search strategy was adapted to fit the formatting and syntax requirements of each database. As noted above, studies were included if they focused on interruption or disengagement and subsequent tracing of clients with the intention to return them to care, rather than solely for the purpose of documenting post-interruption outcomes.

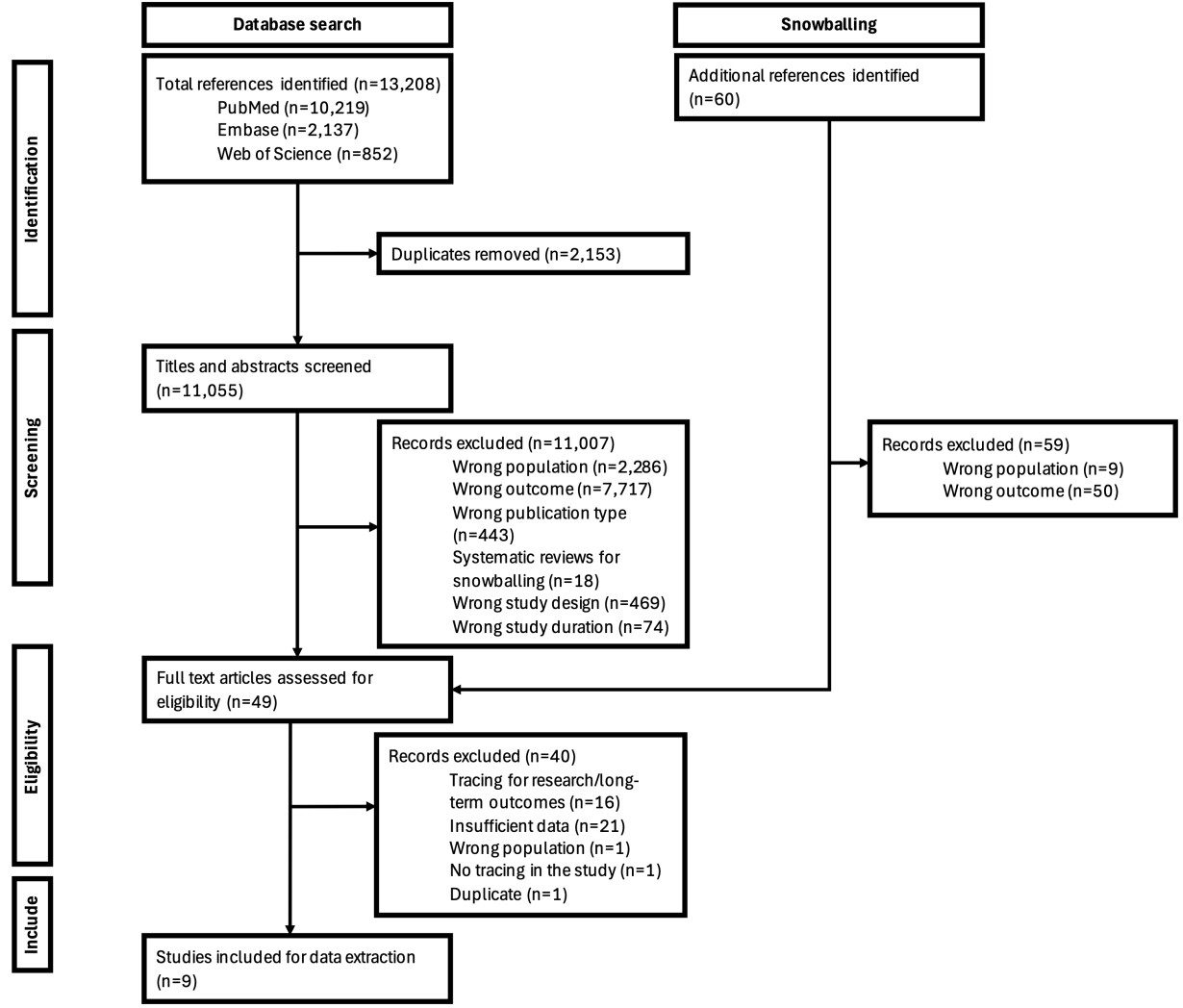

**Fig 1. Tracing cascade and reporting framework.**

Reference lists of relevant studies and review articles were also screened for additional eligible publications. Search terms are shown in S2 Text.

We also searched abstracts from the International AIDS Society Abstract Archive, which includes abstracts presented at AIDS, IAS, and other conferences from 2006-2025; the Conference on Retroviruses and Opportunistic Infections (CROI) Abstract Database, which contains abstracts from 2014-2025; and program books from the International Conference on AIDS and STIs in Africa (ICASA), a biannual conference, from 2011-2023. The search strategy for abstracts used key search terms for both abstract databases and abstract books, based on the search capabilities available from each conference.

Specific inclusion and exclusion criteria are shown in Table 1.

Three coauthors independently screened the abstracts, with each abstract reviewed by at least two authors and discrepancies resolved through consensus. Full-text screening followed the same approach, conducted by two

**Table 1. Inclusion and exclusion criteria.**

| Parameter | Inclusion criteria | Exclusion criteria |
|---|---|---|
| Population | Adults (age 18 years or older); HIV positive; previous disengagement from antiretroviral treatment; targeted for re-engagement through a tracing intervention (i.e., individual met study's criteria for tracing and tracing was attempted; whether or not actual tracing contact was made was not a criterion). | Pediatric/adolescent clients under the age of 18; pregnant/postpartum clients who are receiving HIV services through an ANC clinic or through an Option B+ regimen; currently on ART for HIV prevention (PrEP). |
| Geographic region | Sub-Saharan Africa. | All other regions. |
| Intervention | Tracing interventions in which healthcare personnel (including community health workers, lay counselors, and volunteers) actively follow up clients who have disengaged from ART with the aim to re-engage them in care at the same facility. Follow up may be electronic or in person, by any cadre of healthcare worker. | Clients who were traced by study or clinical staff for the exclusive purpose of determining outcomes after interruption, without attempts to return the clients to care. |
| Study design | Original research articles, including cohort studies, intervention evaluations, observational studies, clinical trials. | Qualitative research; protocols; case study/series; mathematical models; commentaries. |
| Desired descriptive data | Description of tracing activity; criteria for triggering tracing; proportion of target clients reached by tracing intervention; proportion of clients re-engaged after tracing intervention; client sociodemographic information (age, gender, socio-economic status, HIV history/treatment, etc.). | Lacked information about characteristics of those successfully traced. |
| Comparator | Clients targeted by tracing interventions (actual or attempted tracing) but not re-engaged in care. | n.a. |
| Outcomes | Client characteristics by return status following a tracing intervention. | Lacked information about differences between those successfully and unsuccessfully re-engaged in care after tracing. |
| Timing | Majority of data generated between January 1, 2004 and July 1, 2025. | Majority of data generated prior to January 1, 2004 or after July 1, 2025. |

coauthors. Once study eligibility was confirmed, one author extracted the data, which was then verified by a second author.

The following data fields were extracted for each included study: 1) country and setting; 2) study design; 3) total relevant population [Fig 1, box 1] (overall sample size by age and sex where indicated); 4) disengaged population eligible for the tracing intervention [Fig 1, box 2a] (proportion of overall sample, definitions, age and sex where indicated); 5) tracing details (period, time since LTFU, methods: phone calls, home visits, or both; tracing personnel; counseling or encouragement; proportion re-engaged); 6) characteristics of those who returned to care [Fig 1, box 4a] or remained disengaged after tracing or tracing attempt [Fig 1, box 4b] (e.g., age, sex, rural/urban setting, distance from clinic, socio-economic status, HIV treatment history, household characteristics, behavioral characteristics, and/or self-reported preferences and concerns). Descriptive statistics and/or measures of association were extracted for each of these variables, where possible. Discrepancies were resolved through discussion among the authors, with any unresolved cases referred to a third author for final decision-making.

We note that for this review, we did not include "control" or "comparison" populations from comparative studies where these populations did not receive the tracing intervention—our interest was solely in the populations who were eligible for each study's intervention.

## Data analysis

The outcomes of interest for this review were characteristics that described participants who returned to care following a tracing intervention or tracing attempt [Fig 1, box 4a], compared to those who did not [Fig 1, box 4b]. We accepted each

publication's own definitions for identifying ART clients who required tracing (interrupted, disengaged, lost to follow up), tracing intervention, and decisions as to whether an individual returned to care within a specified time period that could be associated with the tracing intervention. Our population of interest included all those eligible for the tracing intervention reported by each study [Fig 1, box 2a], not solely those who were able to be contacted [Fig 1, box 3a].

To structure the results of our search, we used a spreadsheet to compile the extracted data with one row per study to allow for comparing and contrasting across studies (S1 Table). While we prioritized characteristics aligned with our review objectives, such as age, sex, location, socio-economic status, and HIV treatment history, we included all participant characteristics reported in each paper and intervention-specific characteristics such as the intended time between most recent interaction and tracing effort. We report proportions of participants with each characteristic as stated in the original publications, making and reporting adjustments as necessary to allow for consistent comparisons. We also report any further analysis conducted, such as stratifications or estimates of relative risks by characteristic, if presented.

Where possible, we distinguished characteristics among three groups: reached by tracing and returned [Fig 1, box 4a]; reached by tracing but did not return [Fig 1, box 4b]; and not reached by tracing (no contact made [Fig 1, box 3b]), with the sum of 4a and 4b equal to box 3a. We then looked for common characteristics of the groups that could be reported in aggregate across more than one study. Aggregating results across multiple papers masks differences in the specific tracing interventions utilized, and as such, in this aggregate analysis we regarded all interventions as a single tracing approach. Characteristics or outcomes not reported in the original studies were recorded as "not reported (NR)" and no imputation or inference was performed (S1 Table).

Study quality was assessed using the Johanna Briggs Institute (JBI) critical appraisal tools to evaluate the alignment of each included study with the requirements of this systematic review question [26]. JBI provides specific checklists tailored to different study designs. For our assessment, we used the checklists for analytical cross-sectional studies, cohort studies, and randomized controlled trials [26]. The assessments are comprised of 8, 11, and 13 items, respectively, assessing domains related to sampling, measurement of outcomes and/or exposures, and analysis. Each item could be answered as "yes", "no", "unclear", or "not applicable", with "yes" indicating no concerns for that domain. Each study was independently evaluated by two authors. Discrepancies were resolved through consensus. For each tool, the number of "yes" responses was totaled and divided by the total number of applicable items (i.e., questions answered). Articles meeting >70% of the critical appraisal criteria were classified as low risk (good alignment with our review question), those meeting 50–70% were classified as moderate risk, and articles meeting <50% were classified as high risk.

### Ethics

This study utilized only published sources and did not have human subjects research data. No ethics review was therefore required.

## Results

### Search results

Our primary search identified 13,208 unique articles, supplemented by 60 additional articles from secondary searches (Fig 2 and S1 Checklist). Our review of conference abstract books identified 73 relevant abstracts. This large number of search results is likely due to the lack of precise search terms available for the specific intervention of interest. The vast majority of sources identified by our primary search did not report our outcome of interest (for example, many only reported participants' vital status after tracing, not whether they returned to care). After screening titles and abstracts, 49 full-text articles were assessed for eligibility, and nine studies met the inclusion criteria for data extraction. No conference abstracts met the inclusion criteria. A numbered table of all identified studies with reasons for exclusion are provided in S2 Table. Four of the eligible studies included both people living with HIV (PLHIV) who had initiated ART and those who

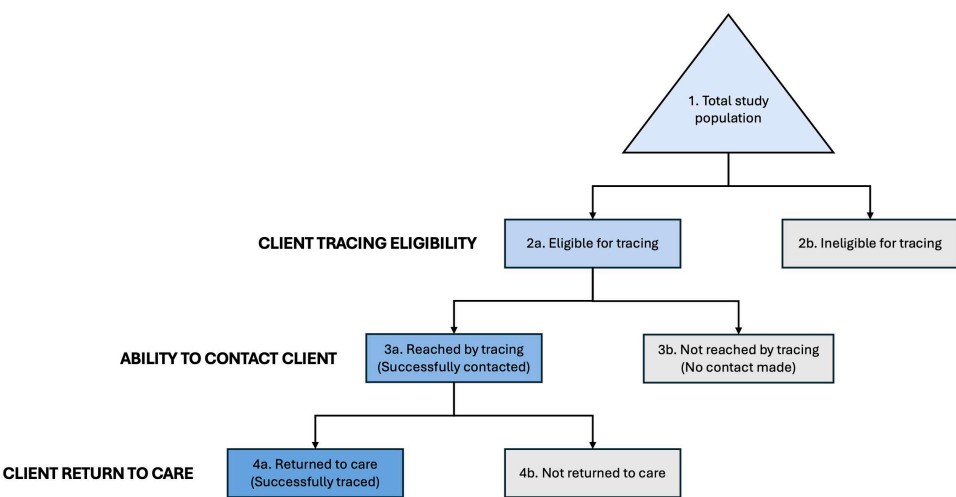

**Fig 2. Preferred reporting items for systematic reviews and meta-analyses (PRISMA) flowchart.**

had not. Since all four provided return to care results stratified by ART status, and the majority of participants in all of these studies had initiated ART, they were included in the final set of articles. Across the nine studies, tracing was attempted for 51,893 clients lost to follow-up (LTFU) [Fig 1, box 2a], of whom 72% (37,544) had initiated ART prior to disengagement.

The nine studies included in the review contributed data from six countries (Table 2), including three from Zambia, two from Malawi, and one each from Kenya, South Africa, and Uganda; one multi-country combined study spanned 14 clinics across Uganda, Kenya, and Tanzania. Each study is assigned a unique code in Table 2, which will be used to refer to the study throughout the rest of this paper.

## Tracing procedures

Criteria for identifying clients eligible for tracing [Fig 1, box 2a] varied significantly across studies (Table 3). Seven studies defined tracing eligibility based on the length of time a client had been disengaged from care, ranging from 21 days (Malawi 1) or 28 days (South Africa 1) to 90 days since last scheduled appointment (Kenya 1, Uganda 1, Zambia 1, Zambia 2, Multi 1). Zambia 3 had the strictest criteria, initiating client tracing just one week after a missed visit. Malawi 2 defined LTFU as missed clinical review or drug refill at intervals of 2, 4, and 6 weeks after a missed visit, with different tracing procedures employed before and after 6 weeks of disengagement.

Tracing activities were conducted by community health workers (CHWs) and peer educators in two studies each and by a range of other personnel in the remaining studies, including outreach workers, designated clinic staff, trained client outreach staff, and lay counselors. Phone calls and home visits were the primary tracing methods in six studies, typically starting with phone calls, with subsequent home visits if phone contact was unsuccessful. Three studies (Kenya 1, Malawi 2, Multi 1) relied exclusively on home visits. Seven studies explicitly reported including counseling or encouragement to return to care in the tracing intervention.

South Africa 1 differed from other studies by employing an mHealth approach in addition to standard tracing procedures. The study utilized public holidays as temporal landmarks to send structured text messages to clients who had disengaged, encouraging them to embrace a 'fresh start' and reinitiate care.

Table 3 also presents tracing outcomes: the proportion of clients attempted to be reached [Fig 1, box 3a and 3b] and the proportion who returned to care [Fig 1, box 4a]. We note that this should not be regarded as a comparison among the

**Table 2. Studies included in the review.**

| Study code | Source | Country | Setting | Dates of tracing intervention | Study design | Sample size; study population* | Sex (% female) | Age (Median (IQR)) or as shown | Comments |
|---|---|---|---|---|---|---|---|---|---|
| Kenya 1 | Rebeiro (2017) [27] | Kenya | 19 AMPATH (Academic Model Providing Access to Healthcare) clinics in western Kenya (largely rural) | 2001 - 2011 (assumed majority of data fall within inclusion criterion 2004 or later) | Routine retrospective record review and data from tracing intervention; some data prior to 2004 | 30,931; clients who were lost to care and not pre-determined to be deaths or transfers (clients pre-determined to be deaths or transfers were not traced) | 68.2 | 35.9 (29.8-43.2) | 45% 13,959/30,931) of sample had not started ART; Results could not be stratified by ART status |
| Malawi 1 | Tweya (2010) [28] | Malawi | 2 ART clinics in Lilongwe (urban) | April 2006 - April 2009 | Prospective cohort | 2,653 clients who missed 3,098 scheduled visits by ≥3 weeks | 55 | 33 | 0.5% (14/2,653) had not started ART; 6% (163/2.653) were children under 14 years; proportions use all missed visits (cases) as the denominator |
| Malawi 2 | Suffrin (2024) [9] | Malawi | A district hospital, a community hospital, and 12 healthcare centers in Neno District (rural) | January 2018 - June 2019 | Retrospective review of data from Tracking Retention And Care Enrollment (TRACE) program | 1,028; clients who were ≥6 weeks late for last scheduled visit | 57 | 30 (23-38) | Sample is limited to participants who were reached by tracer; unclear how many could not be reached |
| South Africa 1† | Njuguna (2024) [29] | South Africa | 24 public health sector priority clinics, Capricorn district, Limpopo province (rural) | 16th June - 18th July 2022 | Three arm individual-level randomized controlled trial | 9,630; participants randomized to 3 arms: standard of care; framed text message (mentioning temporal landmark and fresh start); unframed text message (encouraging return to care) | 69 (Youth Day cohort); 69 (Mandela Day cohort) | 77% between 25 and 49 years for both days | SOC arm was common for both cohorts; YD = Youth Day, MD = Mandela Day |
| Uganda 1 | Nabaggala (2018) [30] | Uganda | Moroto Regional Referral Hospital, North-Eastern Uganda (rural) | January 2014 - August 2015 | Retrospective cohort | 381 PLHIV who had ever missed a scheduled clinic visit | 68 | 30 (23-35) | 20% (75/381) of sample were ART naïve; RTC outcomes stratified by ART status |
| Zambia 1 | Beres (1) (2021) [31] | Zambia | 31 public health facilities across 4 provinces: Eastern, Lusaka, Southern, Western (urban and rural) | 23rd September 2015–17th July 2016 | Nested prospective cohort | 556 LTFU clients confirmed disengaged and surveyed after tracing | 58.3 | 33.6 (28.4-39.9) | This is a nested cohort under Zambia 2; Randomly selected sample of 10% of all LTFU clients at the facilities |
| Zambia 2 | Beres (2) (2021) [32] | Zambia | 71 public health facilities across 4 provinces: Eastern, Lusaka, Southern, Western (urban and rural) | 23rd September 2015–17th July 2016 | Instrumental variable (IV) analysis of a prospective cohort | 4,380 randomized to tracing | 60.4 | 70% between 25 and 44 | |

*(Continued)*

**Table 2.** (Continued)

| Study code | Source | Country | Setting | Dates of tracing intervention | Study design | Sample size; study population* | Sex (% female) | Age (Median (IQR)) or as shown | Comments |
|---|---|---|---|---|---|---|---|---|---|
| Zambia 3 | Krebs (2008) [33] | Zambia | 12 public ART clinics in Lusaka (urban) | May - September 2005 | Open cohort evaluation | 1,343 ART program clients who missed a scheduled visit by >1 week | 62 | 32 (27-39) | 22% (301/1,343) of sample ineligible for ART; RTC outcomes stratified by ART status |
| Multi 1 | Bershetyn (2017) [34] | Uganda, Kenya, Tanzania | 14 clinic sites in Eastern Africa: Mbarara, Uganda; Eldoret, Kenya; Kisumu, Kenya; Kampala, Uganda; Morogoro, Tanzania | 10th June 2011–27th August 2012 [28] | Quasi-experimental study using randomized tracing selection and IV analysis | 991 ART clients >90 days late for their last appointment randomly selected for tracing | 62 | 34 (28-41) | Female clients were stratified by pregnancy status |

*All participants assumed to have interrupted or disengaged from treatment unless stated otherwise.

†Study included further stratification within arms based on specific timing of text messages.

studies, however, as many clients who were not traced may have also returned to care. Proportions returning to care without tracing are generally not reported outside of clinical trials. Four studies compared populations receiving any tracing with no tracing. Zambia 2 and South Africa 1 found no difference in return rates between the groups, while Zambia 3 and Multi 1 observed a modest association with return to care in the traced group.

The reported effectiveness of the studies included in Table 3—the proportion of individuals eligible for tracing who were both reached by tracers and returned to care— varies widely, from a low of 5.4% to a high of 96%, in keeping with the range reported in the Introduction section. Table 3 explains this large range by indicating who is included in each paper's numerator and denominator. The study that reported a 96% success rate, for example (Malawi 2), only reported on participants who were reached by the tracer [Fig 1, box 3a] (hence the 100% contact rate), and the return proportion "excludes an unknown number who could not be found by the tracer." This contrasts with Zambia 3, for example, where 32% of clients eligible for tracing were successfully contacted [Fig 1, box 3a], and 15.4% of the cohort returned to care after tracing [Fig 1, box 4a]. Approaches to tracing differed widely by study, as noted above; Table 3 also makes clear that reporting conventions are equally diverse, making comparison across studies very difficult.

## Characteristics of successfully traced clients [Fig 1, box 4a]

All studies utilized statistical measures such as hazard ratios (HR), odds ratios (OR), and risk ratios (RR) to examine factors associated with return to care after tracing. Only three, however--Malawi 2, Uganda 1, and Zambia 3--compared the characteristics of those who returned to care after tracing [Fig 1, box 4a] to those who were traced but did not return [Fig 1, box 4b].

Age and sex were the most frequently analyzed predictors (Table 4). Five studies found no association between either variable and return to care. Three studies identified significant associations with age. Kenya 1 and the Youth Day messaging group of South Africa 1 reported that older age at ART initiation was associated with a higher likelihood of return compared to younger age, while Malawi 1 found a bimodal association (aOR 2.35 for >39 vs 15–39 and 1.94 for <15 vs 15–39). Three studies identified significant associations with female sex and return to care.

PLOS Global Public Health

**Table 3. Criteria for and methods and results of tracing interventions reported in the studies.**

| Study Code | Study LTFU or disengagement definition | Timing of tracing | Tracing methods | | | Tracers | Proportion of all cases† reached or found (contacted) by tracing intervention [Fig 1, box 3a] | Proportion of all cases† reached and returned to care* [Fig 1, box 4a] |
|---|---|---|---|---|---|---|---|---|
| | | | Phone call | Home visit | Counseling or encouragement | | | |
| Kenya 1 | ≥3 months late for visit and found not to have died or transferred | 1-100 days after being flagged as LTFU | No | Yes; no details provided | "Patients traced and found alive are given counseling and encouragement to return to care" | Trained client outreach staff in the community | 45.8% reached within 8 days of being flagged as LTFU (14,178/30,931) | 66.4% (20,524/30,931) |
| Malawi 1 | ≥3 weeks late for next scheduled visit | Within one week of being identified as ≥3 weeks late | Yes, up to a total of 3 calls and/or home visits | Yes, up to a total of 3 calls and/or home visits | "All patients found alive who had not transferred to other ART facilities were asked to return to the clinic and among those agreeing to return a specific appointment was scheduled with the clinic" | Health extension workers, receptionist | 72.7% (2253/3,098) | 24.7% (764/3,098) |
| Malawi 2 | Clients with missed appointments for a clinical review or for a drug refill for 2, 4, or 6 weeks | Biweekly tracing lists created and followed up | No | Yes, up to 3 home visits | "Time is given to patients to share their concerns and/or challenges and to discuss possible solutions with the tracer" followed by referral for services | Community health workers and study-trained lay, non-clinical staff | 100% (only those traced were included) | 96% (982/1,028); excludes unknown number who could not be found by tracer |
| South Africa 1 | Missed last scheduled clinic appointment by >28 days | For text message arms: Two messages, 5 days before and 1 day after landmark; All arms: telephone attempts within 21 days of missed visit | Yes, 3 attempts within 21 days of missed visit | Yes, if phone call(s) unsuccessful | Two encouragement messages in SMS form, with or without "fresh start" framing | Ward-based primary healthcare outreach teams for all arms; Text messages for text message arms | Message delivery rates: 51–56% across all text message arms | YD: SOC: 11.9%; Framed text message: 12.3%; Unframed text message: 14.5%; MD: SOC: 5.4%; Framed text message: 6.7%; Unframed text message: 6.8% |
| Uganda 1 | Failure to return to care by 90 days of after a missed scheduled visit | 5 days after missed visit up to 90 days | Yes, up to 3 attempts | Yes, if phone call(s) unsuccessful | Encouragement to return to active follow-up and addressing psychosocial issues | Designated counsellor, clinic staff, expert clients, peer leaders | 85% (598 phone calls and 472 home visits in total) | 70% (267/381) |
| Zambia 1 | >90 days late for an ART appointment with no evidence of transfer or death | Not explicitly reported | Yes | Yes, if phone call unsuccessful | Yes, verbally encouraged to return to care | Trained peer educators | 100% (only those found were included) | 73% (406/556) |

*(Continued)*

**Table 3.** (Continued)

| Study Code | Study LTFU or disengagement definition | Timing of tracing | Tracing methods | | | Tracers | Proportion of all cases† reached or found (contacted) by tracing intervention [Fig 1, box 3a] | Proportion of all cases† reached and returned to care* [Fig 1, box 4a] |
|---|---|---|---|---|---|---|---|---|
| | | | Phone call | Home visit | Counseling or encouragement | | | |
| Zambia 2 | >90 days late from last scheduled appointment between Aug 2013–July 2015, with unknown status | Not explicitly reported | Yes | Yes, at least 3 in-person visit attempts | Yes, verbally encouraged to return to care; Some tracers accompanied clients to facility | Trained peer educators | 26.4% (1158/4380) | 26.7% (1,168/4,380) among those randomized to tracing vs 25.1% (4,973/19,784) not randomized to tracing (HR: 1.06, 95% CI (1.00–1.13), p-value = 0.06) |
| Zambia 3 | Missed scheduled visit by > 1 week | Varied across sites | No | Yes | Yes, reminders and encouragement provided | Home-based caregivers | 32% (430/1343) | 15.4% (207/1,343) of all LTFU; 31% (133/430) of those traced (aRR: 2.3; 95% CI (1.7, 3.2)) |
| Multi 1 | >3 months late to a scheduled appointment or not seen for 4 months if no follow-up was scheduled | Varied across sites | Not reported | Yes | Yes, offered during semi structured interviews | Community health workers | 36% (360/991) contacted at home | 13.3% of traced vs. 10.0% of not traced returned (aHR = 1.30; 95% CI (1.08, 1.58); p = 0.006) |

LTFU: lost to follow up; MD: Mandela Day (holiday); SOC: standard of care; YD: Youth Day (holiday); AHR: adjusted hazard ratio.

*Includes all who met study eligibility criteria for the tracing intervention, whether or not they were actually contacted by a tracer.

†Cases: either indicate total clients lost to follow-up or total missed appointments (Malawi 1).

In addition to age and sex, most of the papers reported at least one or more additional significant predictors of returning to care following a tracing intervention, though there was little consistency in which variables were analyzed across studies (Table 5). Predictors spanned a wide range of social, behavioral, and structural characteristics.

The timing of tracing relative to a client's last healthcare system interaction (clinic visit or medication pickup) emerged as a common determinant. Three studies found a positive association between early tracing and return to care. One study reported that tracing within 8 days of a missed visit was twice as likely to result in a return to care (Kenya 1), while another reported a steep drop-off in return rates after two weeks (Multi 1). Similarly, South Africa 1 observed significantly lower return rates when tracing occurred after 6 months of interruption compared to before 6 months. Contrary to these studies, Zambia 2 found a greater benefit when tracing occurred >6 months after LTFU than within 3 months.

Socioeconomic and psychosocial factors were inconsistently analyzed. Zambia 1 found that clients who confronted a stigmatizer once in the last 12 months or who were contacted more than the standard three times by the healthcare facility after a previous missed visit had a higher likelihood of return. It also found that clients receiving care from urban facilities or hospital settings, compared to rural facilities, were less likely to return. Kenya 1 found slightly higher return rates for clients who disclosed their HIV status.

**Table 4. Sex and age as predictors of returning to care after tracing intervention.**

| Study code | Age as a predictor of return to care (return) | Sex as a predictor of return to care (return) |
|---|---|---|
| Kenya 1 | Older age associated with return<br>• HR: 1.21 for clients age 25-34.9 (compared to 18-24.9); 95% CI (1.14, 1.29)<br>• HR: 1.39 for clients age 35-44.9 (compared to 18-24.9); 95% CI (1.31, 1.48)<br>• HR: 1.52 for clients age ≥ 45 (compared to 18-24.9); 95% CI (1.42, 1.63) | Female sex associated with return<br>• HR: 1.05 for non-pregnant women (compared to men); 95% CI (1.02, 1.09)<br>• HR: 1.17 for pregnant women (compared to men); 95% CI (1.10, 1.26) |
| Malawi 1 | Older age associated with return<br>• aOR: 2.35 for clients aged ≥40 at ART initiation, compared to those aged 15–39; 95% CI (1.51, 3.64) | Female sex associated with return<br>• aOR: 0.56 for men (compared to women); 95% CI (0.39, 0.80) |
| Malawi 2 | Age not associated with return<br>• 91-97% of clients returned in all age groups, difference not significant<br>• aOR: 0.30 for clients aged 15–29 (compared <15 years); 95% CI (0.05-1.44)<br>• aOR: 0.27 for clients aged 30–39 (compared <15 years); 95% CI (0.18-1.23)<br>• aOR: 0.66 for clients aged 40–49 (compared <15 years); 95% CI (0.11-4.04)<br>• aOR: 0.41 for clients aged >50 (compared <15 years); 95% CI (0.06-2.74) | Sex not associated with return<br>• aOR: 0.85 for men (compared to women); 95% CI (0.50, 1.47) |
| South Africa 1 | Older age associated with return<br>• YD – aOR: 1.65 for clients who initiated treatment at age > 50, compared to those aged 18–24; 95% CI (1.09, 2.49)<br>Age not associated with return<br>• MD - No significant difference by age group in probability of return; aOR: 0.83 for >50, 95% CI (0.50, 1.36) | Sex not associated with return<br>• YD – aOR: 1.03 for females (compared to men); 95% CI (0.85, 1.25)<br>• MD – aOR: 1.23 for females (compared to men); 95% CI (0.94, 1.60) |
| Uganda 1 | Age not associated with return<br>• aOR: 0.99; 95% CI (0.98–1.01) | Female sex associated with return<br>• aOR: 1.23 for women (compared to men); 95% CI (1.05–1.43) |
| Zambia 1 | Age not associated with return<br>• aHR: 1.21 for clients age ≥ 45, compared to 18–24 years; 95% CI (0.68–2.17) | Sex not associated with return<br>• aHR 0.91 for men (compared to women); 95% CI (0.65–1.26) |
| Zambia 2 | Not reported | Not reported |
| Zambia 3 | Age not associated with return | Sex not associated with return |
| Multi 1 | Not reported | Not reported |

HR: hazard ratio; aHR: adjusted hazard ratio; OR: odds ratio; aOR: adjusted odds ratio; 95% CI: 95% confidence intervals; YD: Youth Day; MD: Mandela Day.

Clinical factors were also inconsistently reported, suggesting mixed associations with return to care. With regard to WHO staging, Malawi 1 reported that female clients with WHO stage 4 had higher odds of return, and Malawi 2 reported lower odds for male clients with WHO stage 4. While for male clients in Malawi 1, and all clients in Kenya 1, Zambia 3, and Multi 1, there was no significant association between staging and return to care.

Most studies did not find significant associations with CD4 counts and return to care (Zambia 1, Zambia 2, and Multi 1); one study found higher CD4 counts to be associated with a lower likelihood of return (Kenya 1). Among the three studies that enrolled both ART and non-ART populations, being on ART at the time of tracing was found to be a significant positive predictor of return in Kenya 1 and Zambia 3, but no association was established in Uganda 1. Malawi 1 only had 0.5% of non-initiators and did not explore this association.

## Study quality assessment

Of the nine studies included in this review, there were two cross-sectional studies, five cohort studies, and one randomized controlled trial. All studies were appraised as having low or moderate risk according to the criteria described above, with scores ranging from 73-88% for four studies at low risk (good alignment with our review question) and 62–69% for five studies at moderate risk. S3 Table provides details of the quality assessment. Although methodological limitations

PLOS Global Public Health

**Table 5. Other predictors of returning to care after tracing intervention.**

| Study code | Time interval between LTFU and tracing intervention | Previous interruptions to care | Factors influencing return to care (return) after tracing intervention (whether contacted or not) | Proportion traced who were still on ART* |
|---|---|---|---|---|
| Kenya 1 | • For each log time unit delay, hazard of return reduced by 14% (aHR: 0.86; 95% CI (0.85, 0.88)) <br>• Early contact (within 8 days of missed visit) associated with return compared to never reached/late contact (HR: 2.06; 95% CI (1.99, 2.11)) <br>• Effect of tracing diminished by time since missed visit (HR: 2.50 within 8 days) (HR: <1.30 after 18 months) | Not reported | • Successful outreach (HR: 1.43; 95% CI (1.34, 1.53); p<0.001) <br>• On ART (HR: 1.60; 95% CI (1.54, 1.66); p<0.001) <br>• HIV status disclosure (HR: 1.06; 95% CI (1.02, 1.09); p<0.001) <br>• CD4 at enrollment ≥350 (HR: 0.95; 95% CI (0.91, 0.99); p=0.016) | 2.4% (formal transfers only) |
| Malawi 1 | Not reported | • Self-reported treatment gap (missed doses) prior to tracing interaction associated with return to care (aOR: 8.69; 95% CI (3.92, 19.29)) | • Women WHO stage IV at ART initiation (aOR: 2.14; 95% CI (1.07, 4.25)) <br>• Women >6 months on ART (aOR: 2.10; 95% CI (1.29, 3.43)) <br>• Uninterrupted therapy (aOR: 3.11 (2.01, 4.83) <br>• Inaccurate locator information (26% could not be traced) | 46% (Collected ART from alternative sources (21%) or transferred to another clinic (25%) |
| Malawi 2 | Not reported | Not reported | • Men WHO stage IV (associated with low 6-month return) (aOR: 0.18; 95% CI (0.06, 0.54); p=0.002) <br>• Men WHO stage IV (associated with low 12-month return) (aOR: 0.12; 95% CI (0.04, 0.16); p=0.002) <br>• Initiation after 2016 (associated with low 6-month return) (aOR: 0.08; 95% CI (0.06, 0.18); p<0.001) <br>• Initiation after 2016 (associated with low 12-month return) (aOR: 0.08; 95% CI (0.04, 0.16); p<0.001) <br>• Initiation after 2016 (associated with low 24-month return) (aOR: 0.16; 95% CI (0.10, 0.25); p<0.001) | Not reported |
| South Africa 1 | • YD: Treatment interruption ≥6 months associated with reduced odds of return (aOR: 0.04; 95% CI (0.03, 0.05); p=0.000) <br>• MD: Treatment interruption ≥6 months associated with reduced odds of return (aOR: 0.14; 95% CI (0.11, 0.19); p=0.000) | Not reported | • YD: 6–12 months on ART (aOR: 2.43; 95% CI (1.58, 3.74); p=0.000) <br>• YD: >12 months on ART (aOR: 3.21; 95% CI (2.35, 4.38); p=0.000) <br>• MD: >12 months on ART (aOR: 1.96; 95% CI (1.36, 2.83); p=0.000) <br>• MD: Enrollment in differentiated care (aOR: 1.59; 95% CI (1.21, 2.10); p=0.001) <br>• YD: Enrollment in designated priority clinic (DOH high-volume site) associated with reduced return (aOR 0.72; 95% CI (0.60, 0.87); p=0.001) <br>• MD: Enrollment priority clinic had reduced return probability (aOR 0.71; 95% CI (0.54, 0.92); p=0.010) <br>• Those who received any text message had increased but nonsignificant odds of return compared to no text (YD aOR: 1.17; 95% CI (0.98, 1.40); p=0.07) (MD aOR:1.22; 95% CI (0.96, 1.55); p=1.0) | 4.5% |
| Uganda 1 | Not reported | Not reported | • Other reasons for missed appointment such as long distance to clinic, stigma, extra drug supplies, forgetting compared to travel as a reason for missed appointment (aRR: 0.41; 95% CI (0.28, 0.59); p<0.001) <br>• Unavailable/unreachable at the time of contact compared to travel as a reason for missed appointment (aRR: 0.85; 95% CI (0.75, 0.95); p<0.01) <br>• No significant association between ART status (aRR for on ART vs ART-naive: 1.16; 95% CI (0.96, 1.39); p<0.12) or tracking method (aRR for home visit vs phone call: 0.98; 95% CI (0.87, 1.11); p<0.83) and return to care | Not reported |

*(Continued)*

**Table 5.** (Continued)

| Study code | Time interval between LTFU and tracing intervention | Previous interruptions to care | Factors influencing return to care (return) after tracing intervention (whether contacted or not) | Proportion traced who were still on ART* |
|---|---|---|---|---|
| Zambia 1 | • >18 months from HIV care enrolment to disengagement associated with reduced odds of return (compared to ≤18 months from enrolment to disengagement) (aHR: 0.58, 95% CI (0.36, 0.94)) | • Experiencing previous gap in care associated with return (aHR: 1.95; 95% CI (1.23, 3.09)) | • Confronting stigmatizer one time in past 12 months (compared to never) (aHR: 2.14; 95% CI (1.25, 3.65)) <br>• Being contacted >3 times by facility at a previous missed visit (compared to never) (aHR: 2.65; 95% CI (1.04, 6.73)) <br>• Clients from urban health centers, lower hazard of return (compared to rural health center) (aHR: 0.68; 95% CI (0.48–0.96)) <br>• Clients from hospitals, lower hazard of return (compared to rural health center) (aHR: 0.52; 95% CI (0.33, 0.82)) <br>• Highest wealth tertile associated with lower return (compared to lowest tertile) (aHR: 0.71; 95% CI (0.47, 1.18) <br>• Middle wealth tertile associated with increased return (compared to lowest tertile) (aHR: 1.27; 95% CI (0.89, 1.80)) | 0% (This population was confirmed to be disengaged from care) |
| Zambia 2 | • Longer time since LTFU associated with greater benefit. Effect was stronger when tracing occurred >6 months after LTFU (ARD: +13%, 95% CI (6–20%)) vs. <3 months (ARD: –21%, 95% CI (–42% to –1%)) | Not reported | • Early return to care following tracing (highest IR of return was within 1 week after tracer contact: IR 5.74, 95% CI (3.78, 8.71)) vs 2 weeks-1month > subsequent periods) <br>• No significant association between randomization to tracing and return to care (HR: 1.06, 95% CI (1.00–1.13), p-value = 0.06) | At least 37% (1621/4380) (authors' calculation) |
| Zambia 3 | Not reported | Not reported | • Being on ART was associated with return (88% vs. 12% compared to those not on ART; p < 0.0001) <br>• Tracing associated with higher likelihood of return (aRR: 2.3, 95% CI (1.7, 3.2)) | Not reported properly; 4% (17/430) reported surplus meds |
| Multi 1 | • Not directly reported; Return rates were highest in the first 2 weeks (aHR: 10.5, 95% CI (5.4, 20.1)) after tracing than 2 weeks to 3 months (aHR: 2.4, 95% CI (1.1, 5.1)) and 3–6 months (aHR: 2.8, 95% CI (1.0, 8.1)) vs no tracing. <br>• Tracing effect had a half-life of ~7 days. Return rates rose sharply in 0–14 days post-tracing then declined. | Not reported | • Out of care clients who were contacted in person (RR: 2.47, 95% CI (1.05, 5.78); aRD: 22.1%, 95% CI (7.1%, 36.2%) vs those not contacted) <br>• Tracing vs no tracing (aHR: 1.30, 95% CI (1.08, 1.58)) | 21.4% (212/991); These numbers are for the 360/991 contacted in-person |

HR: hazard ratio; aHR: adjusted hazard ratio; RR: risk ratio; aOR: adjusted odds ratio; aRD: adjusted risk difference; IR: Incidence rate; 95% CI: 95% confidence intervals.

*At the same facility; transferred to another facility; or had other ARV access such as sharing with partner.

were noted in several domains, particularly in the clarity of denominators, stemming partly from the different definitions of LTFU and differential reporting of outcomes, as well as follow-up procedures, and blinding, most studies demonstrated sufficient alignment with the review question to be considered at relatively low risk overall. The overall quality of studies may be underestimated here, as our appraisal was focused on alignment with our systematic review question rather than the primary exposure or outcome measures used in the original studies.

## Discussion

This systematic review sought client characteristics associated with return to care and identified differences between clients traced and returned to care and those who were contacted (or for whom contact was attempted) but did not return in

sub-Saharan Africa. Positive client-level predictors of return after tracing included older age, female sex, and disclosure of HIV status. Higher CD4 counts and/or lower WHO stages at initiation had mixed associations with return. Earlier initiation of tracing by healthcare providers after recording an interruption was also associated with higher probability of return. The only negative predictor identified was a delay in tracing—the converse of the early tracing that predicted success – with waiting longer to trace clients associated with a reduction in return to care.

The few client characteristics that were associated, though inconsistently, with successful tracing included being older or in stable clinical condition at ART initiation. Older age has been identified as a predictor of retention in care in multiple studies [35–37]. Factors such as increased motivation to protect one's health, HIV status disclosure, stable family and relationship status, and having overcome stigma may also drive older adults to re-engage in care after missing visits [35–37]. Sex also mattered: in three out of seven studies, female clients were more likely to return to care after tracing than male clients. These results are consistent with broader literature that suggests that female and male clients experience HIV testing and treatment differently [35]; it is not surprising that these populations may also interact with tracing interventions differently. Men are often more reluctant to seek care in the first place and are prone to travel for employment or familial duties which may contribute to a higher likelihood of disengagement [35,38,39].

Clients with advanced disease stages and/or low CD4 counts, in contrast, were both less likely to return to care after tracing and more likely to be deceased [9,27–29,31,34]. This underscores a fundamental dilemma in determining how to prioritize responses to treatment interruptions: the clients who are most clinically vulnerable (high-risk groups) may be least likely to benefit from tracing as an intervention, unless it is conducted promptly after disengagement. Identifying other interventions for high-risk clients may thus be critical to improving survival outcomes in this population.

Early tracing, within 8–14 days of a missed appointment, generally resulted in more successful tracing outcomes [27,34]. The optimal timing of tracing is complicated, however, by the fact that a considerable proportion of clients with short interruptions return to care voluntarily, likely prior to viral rebound or development of serious symptoms. Kenya 1, for example, reported that in their cohort, approximately half of patients with a gap in care returned spontaneously within three months, and noted that outreach had the greatest effect when conducted between 1 and 6 months after a gap in care [27]. This paper also mentioned the potential value of optimization modeling to identify the period when tracing is most likely to be successful, in terms of time after last interaction. Our review supports the argument that very late tracing (>6 months after last interaction) is not effective; the frequency of voluntary return within a few days suggests that very early tracing is also sub-optimal. Further research is needed to identify the optimal window for tracing that would maximize the likelihood of return to care. To do this, routine medical record systems for ART clients should record whether and when tracing was attempted, fields that are not currently collected for most countries.

In addition to voluntary returns, many clients identified as disengaged were found to be misclassified—either still in care at the same facility, silently self-transferred to another facility, or continuing ART through informal means such as medication sharing. Malawi 1, for example, found that while 3% of LTFU were silent transfers, 12% had officially transferred but lacked clinic documentation of that fact, and another 13% collected drugs from alternative sources and were still on ART. Another study in Cameroon reported that 19% of LTFU clients were misclassified due to silent transfer (6%), improper documentation (7%), or continuing ART using ARVs from previous visits or informal sources (6%) [40]. As a result, tracing efforts may have been directed toward clients who were not truly disengaged, reducing the efficiency of these interventions. The frequency of silent transfers, which are often difficult to identify in current medical record systems, combined with limited inter-clinic coordination has been reported as a challenge to tracing interventions in other contexts [41]. Improving systems for documenting transfers and deaths could help ensure that tracing targets those genuinely out of care [7,40].

In addition to the challenges in improving the efficiency of tracing as an intervention, our review identified a number of problems with the reporting and evaluation of tracing activities. Our difficulty in describing the subset of clients for whom tracing was successful stemmed in part from the lack of standardized methods and language used, which limited our

ability to make meaningful comparisons across studies. As is typical in the HIV cascade literature [24], definitions of disengagement and outcomes varied widely across the nine studies included in this review. Each used different time thresholds to classify clients as eligible for tracing, ranging from one week to three months. Most also used different definitions of return to care and other outcomes to assess tracing impacts. Certain studies conducted tracing even before designating a client as lost to follow-up. Some studies excluded study clients who were later found to have died or transferred to a new facility from their analysis, on the grounds that these individuals could not "return to care." Others combined clients who had and had not initiated ART. Several did not provide clear definitions of these terms at all.

Future work would benefit from greater precision on the part of researchers in explaining exactly who was included in their studies and by standardizing outcomes. Tracing interventions, for example, might report a standard outcome of "received medication within 30 days of tracing contact," alongside whatever other outcomes are of interest. Since tracing seems likely to persist as a core intervention in many countries' HIV management guidelines (Panel 1), standardizing reporting could help identify ways to increase its effectiveness by facilitating cross-study comparisons. The framework proposed in this review (Fig 1) offers an initial, practical approach by outlining coded outcome categories that could be used consistently to support such comparisons.

Though we could find few patient- or facility-level characteristics consistently associated with tracing success in the published literature, it seems likely that such characteristics exist, given the wide range of results in the studies reported here. Further research to identify who is most likely to return to care due to tracing—the "high-benefit" group—could enhance the overall impact of tracing at a time of severe resource constraints. Current tracing approaches generally include anyone classified as lost to follow up. While this inclusive strategy throws the widest possible net, it is unlikely to make the most effective use of limited resources, particularly in settings where voluntary returns and undocumented transfers are common and nongovernmental partners that previously provided tracing support have lost their funding. Modeling studies have illustrated the value of a high-benefit approach in other contexts and for other diseases [22,42]. ART client tracing may offer another opportunity for this approach to improve intervention efficiency.

By identifying who should be prioritized for tracing, moreover, we also can identify populations for whom tracing is less likely to lead to return to care. In identifying these populations, clinicians and implementers can seek more effective interventions, rather than continuing to invest resources in repeated tracing attempts. Community partnerships and community-based support are one such intervention that have been identified to reduce gaps in care for men and younger clients [35]. Targeted activities for men to encourage continuation of care have proven successful and acceptable. A South African program, for example pairs clients with treatment 'coaches' who encourage them to adhere to care and return if they do disengage. This very personalized tracing may be more successful than standard clinical tracing due to the trusted personal relationships between clients and their coaches [43]. Targeted outreach activities such as men's clubs, community-based peer support initiatives may offer more successful avenues for re-engagement for these populations.

As might be expected from the discussion above, our review had several limitations. First, while we believe that our search of the peer-reviewed, published literature and abstracts was thorough, the lack of standard terminology for describing both disengagement and tracing as an intervention hampered the creation of precise search strings, and it is possible that some sources were missed, particularly experiences reported in the unpublished (gray) literature. Potential inaccuracies related to translation of scientific terminology may have hampered our search. Second, the small number of eligible studies found, limited number of countries represented (six), variability in definitions of LTFU, heterogeneity in tracing approaches employed, and inconsistent reporting of intervention timing, design, and outcomes severely limited the generalizability of our results. Finally, even the eligible papers included in this review varied in how they calculated results, with some explicitly excluding clients who could not be found by tracers, some including clients who never started ART, and others not specifying exactly which participants were reflected in their presentations of client characteristics or in their tracing results.

 

## Conclusions

In view of the results presented here, and the limitations listed above, we conclude that the most important finding of our review is how little is known about the characteristics of individuals for whom tracing leads to re-engagement in care. While there is a large body of literature about tracing interventions overall [7,11,34,44–46] and some evidence about the impact of tracing compared to no intervention [29,31,33,34], very little attention has been paid to the question of which types of clients are more likely to respond positively to tracing and thus be a good choice for using the resources required for this intervention. Without this information, it is difficult to determine whether meaningful differences exist between these groups that could inform more efficient targeting of tracing interventions and improve allocation of limited resources. As countries struggle to sustain this intervention in the face of funding cuts and partner withdrawal, better information about who benefits from tracing after disengagement, rather than solely who is at risk from disengagement itself, has the potential to make tracing programs more effective and identify those who need something other than tracing to return them to care. Future research should also explore predictive models to help identify and prioritize both high-risk and high-benefit clients, for this and other interventions.

## Supporting information

**S1 Text. Prospero protocol.**
(PDF)

**S2 Text. Search terms.**
(DOCX)

**S1 Checklist. PRISMA checklist.**
(PDF)

**S1 Table. Data extraction table for included studies used in the systematic review.**
(XLSX)

**S2 Table. All studies identified in the literature search with reasons of exclusion.**
(XLSX)

**S3 Table. Study quality assessment.**
(DOCX)

**S1 Fig. PRISMA 2020 main checklist.**
(PDF)

## Author contributions

**Conceptualization:** Anushka Marri, Allison Morgan, Mhairi Maskew, Sydney Rosen.

**Data curation:** Anushka Marri, Allison Morgan, David B Flynn, Sydney Rosen.

**Formal analysis:** Anushka Marri, Allison Morgan, Mariet Benade, Mhairi Maskew, Nyasha Mutanda, Sydney Rosen.

**Funding acquisition:** Sydney Rosen.

**Writing – original draft:** Anushka Marri, Allison Morgan, Sydney Rosen.

**Writing – review & editing:** Anushka Marri, Allison Morgan, Mariet Benade, David B Flynn, Mhairi Maskew, Nyasha Mutanda, Sydney Rosen.

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
