## [Decision Letter · Decision Letter 0]

22 Dec 2025

PGPH-D-25-02937

Who does tracing work for? Characteristics of clients successfully re-engaged in ART care in sub-Saharan Africa after a tracing intervention: a systematic review

Dear Dr. Rosen,

Thank you for submitting your manuscript to PLOS Global Public Health. After careful consideration, we feel that it has merit but does not fully meet PLOS Global Public Health’s publication criteria as it currently stands. Therefore, we invite you to submit a revised version of the manuscript that addresses the points raised during the review process.

We look forward to receiving your revised manuscript.

Kind regards,

Lina Taing

Academic Editor

Journal Requirements:

1. We have noticed that you have cited Supplementary Table 3 in your manuscript. However, there are no corresponding files uploaded to the submission. Please upload them as separate files with the item type 'Supporting Information'.

2. As required by our policy on Data Availability, please ensure your manuscript or supplementary information includes the following:

3. We note that your Data Availability Statement is currently as follows: “All data relevant to the study are included in the article or uploaded as online supplemental information.”

Additional Editor Comments (if provided):

This paper makes an important conceptual and practical contribution to HIV care by questioning the conventional "high-risk approach" to tracing prioritization-which focuses on disease severity and transmission risk-and proposing instead a "high-benefit approach" that considers which client characteristics predict successful return to care after tracing. Tracing disengaged clients or those lost to follow up are critical but resource-intensive interventions for ensuring people living with HIV access lifelong treatment. By systematically identifying that demographics such as older age and female sex, along with early tracing, frequent contact attempts, and certain psychosocial factors predict return to care, the authors provide evidence-based guidance for more efficient resource allocation. This is particularly urgent given extensive funding cuts, including USAID's pullout, which make effective targeting of limited resources essential.

The systematic review itself is methodologically rigorous and appropriately scoped. The authors searched major medical and interdisciplinary databases over two decades, as well as archives from the International AIDS Society and ICASA-leading institutions that convene international and African AIDS specialists-focusing on sub-Saharan Africa, which bears the world's largest HIV burden. The design, including PROSPERO registration, demonstrates careful planning. While only nine studies met inclusion criteria for comparing characteristics of clients who returned to care versus those who did not, the authors are admirably transparent about limitations including inconsistent definitions, heterogeneous interventions, and varied reporting. Their call for standardized definitions and outcomes is especially important for the field. Perhaps most critically, the paper identifies a fundamental gap: despite widespread tracing implementation, remarkably little is known about which client subgroups benefit most from this intervention. This finding justifies the review and points toward a concrete implementation science research agenda that could substantially improve tracing program efficiency at a time when such improvements are urgently needed.

This is a well-written manuscript with sound methodology and clear presentation. I recommend two minor revisions to strengthen the work:

• Enhance the abstract by including specific patient characteristics associated with return to care (e.g., age ranges and key demographic details from your findings) to provide readers with immediate insight into which populations are most affected.

• Address the limitations of automated translation tools for scientific terminology, by acknowledging that search engine translators may not accurately render technical language or domain-specific concepts from foreign language sources.

Reviewers' comments:

Reviewer's Responses to Questions

**Comments to the Author**

1. Does this manuscript meet PLOS Global Public Health’s publication criteria?

Reviewer #1: Yes

2. Has the statistical analysis been performed appropriately and rigorously?

Reviewer #1: Yes

3. Have the authors made all data underlying the findings in their manuscript fully available (please refer to the Data Availability Statement at the start of the manuscript PDF file)?

Reviewer #1: Yes

4. Is the manuscript presented in an intelligible fashion and written in standard English?

Reviewer #1: Yes

Reviewer #1: This manuscript describes a systematic review of the literature on tracing of HIV-infected persons in sub-Saharan Africa who discontinue care. Given the high rates of care discontinuation and the considerable resources needed to trace them, this topic is very important to care clinics and policy makers. The authors identified numerous challenges in reviewing the literature, including inconsistent definitions of care discontinuation, diverse terminology, and inconsistent reporting of key variables. They offer a possible framework for improvements in reporting to overcome these shortcomings. Despite these challenges they have identified factors which appear to be associated with successful tracing- time since discontinuation, age and female sex. They have appropriately listed the limitations of their study, and are conservative with their conclusions.

**Do you want your identity to be public for this peer review?** For information about this choice, including consent withdrawal, please see our Privacy Policy

Reviewer #1: **Yes:** John A. Bartlett

---

## [Editor Report · Decision Letter 1]

4 Feb 2026

Who does tracing work for? Characteristics of clients successfully re-engaged in ART care in sub-Saharan Africa after a tracing intervention: a systematic review

PGPH-D-25-02937R1

Dear Professor Rosen,

We are pleased to inform you that your manuscript 'Who does tracing work for? Characteristics of clients successfully re-engaged in ART care in sub-Saharan Africa after a tracing intervention: a systematic review' has been provisionally accepted for publication in PLOS Global Public Health.

Best regards,

Lina Taing

Academic Editor